# A Novel ML Model for Numerical Simulations Leveraging Fourier Neural Operators

**Ali Takbiri-Borujeni**
Amazon
Seattle, WA
alitakb@amazon.com

**Mohammad Kazemi**
Department of Engineering
Slippery Rock University
Slippery Rock, PA, 16057
mo.kazemi@sru.edu

**Sam Takbiri**
University of Isfahan
Isfahan, Iran
sam.takbiri@gmail.com

## Abstract

Numerical simulations for reservoir management for energy recovery optimization involve solving partial differential equations across numerical grids, providing detailed insights into fluid flow, heat transfer, and other critical reservoir behaviors. However, their computational demands often hinder practical implementation due to lengthy runtimes and resource-intensive processes. In this paper, we propose a deep learning methodology to address these challenges. Our approach leverages a neural operator that directly parameterizes the integral kernel in Fourier space. By doing so, we facilitate swift and efficient predictions, effectively reducing the computational burden associated with multiple numerical simulations. The robustness of the proposed approach has been evaluated for simulations of the steam injection process in high-viscosity oil reservoirs, an advanced thermal recovery method used to extract heavy oil from underground reservoirs. Key features of the proposed methodology are that it slashes computational time from hours to seconds, making it feasible for real-time reservoir management decisions. We use input data and corresponding output fields from five numerical models for the steam injection process as training data. This allows the ML model to learn from the complete evolution of the process across diverse simulations. During inference, the ML model relies on the first 10 time steps of numerical simulation results. It then predicts the subsequent 40 time steps in an autoregressive manner, capturing temporal dependencies effectively. The ML model accurately forecasts simulation outcomes at the numerical grid level, with error rates consistently below 10 percent. Beyond reservoir simulation, our approach holds promise for other fields, including computational fluid dynamics (CFD), structural engineering, and weather forecasting.

## 1 Introduction

Constructing numerical models to represent the physics of oil and gas reservoirs is a demanding task, requiring several steps. Initially, a reservoir geometry model is constructed, comprising grid blocks where partial differential equations (PDEs) are solved to determine pressure and oil/gas saturation. Geological attributes, fluid properties, and rock-fluid characteristics serve as inputs, with rock properties being particularly uncertain due to limited measurements Aziz (1979). To validate the reservoir model, a process called history matching is performed Chavent et al. (1975); Kruger et al. (1961); Van Leeuwen (1999); Wahl et al. (1962); Slater & Durrer (1971); Nourozieh et al. (2015), where the model's response is compared against field pressure and fluid flow rate data. Achieving a match involves iteratively adjusting inputs, a laborious task Erbas & Christie (2007). In the reservoir's lifecycle, history matching is performed repeatedly Soares et al. (2021). A history-matched model has to be updated several times during the life of an oil field as we learn more from the reservoir properties, thereby reducing uncertainty as we produce from it and observe recovered fluid properties, flow rates, and pressures at the wellbores.

Various attempts, ranging from gradient-based optimization methods to global optimization approaches, have been made to automate the history matching process and address computational challenges in numerical reservoir simulations Tyler et al. (1993); Van Leeuwen (1999); Evensen

(2003); Haugen et al. (2008); Aanonsen et al. (2009); Szklarz et al. (2011); Rotondi et al. (2006); Erbas & Christie (2007); Maucec et al. (2007). However, the complexity and size of simulation models significantly impact the performance of optimization models for history matching Shahkarami et al. (2018). These challenges have spurred the development of data-driven reservoir simulator models, offering significant reductions in computational time compared to physics-based simulators Tang et al. (2020); Nwachukwu et al. (2018); Zheng et al. (2019); Guo et al. (2018); Zhao et al. (2017); Zhang et al. (2021); Soares et al. (2021); Ghassemzadeh et al. (2021). For instance, proposals have been made for ML-based and statistical reduced-order models (ROMs), also known as response surface models. These models aim to simplify governing equations by reducing the number of variables, inevitably overlooking certain details in the relationships between inputs and outputs. However, limitations exist with these methods. For instance, when designing multiple responses, more response surface models are needed to predict outputs, increasing complexity. Moreover, creating reduced-order models typically requires a large number of simulation runs. Additionally, these methods assume a smooth variation in response based on input parameter continuity, which may not hold for systems with many discrete input parameters, imposing practical limitations.

There are also a handful of ML-based surrogate approaches proposed in the literature. Most of these methods require substantial pre-processing steps (data transformation and feature engineering) for training ML models, and the algorithms used are generally not sophisticated enough to account for shock waves. The proposed approach aims to solve numerical reservoir simulations by leveraging a novel neural operator that parameterizes the integral kernel directly in Fourier space, accelerating predictions by a factor of $10^3$. This significantly cuts down the computational expense, crucial for tasks like history matching, which typically require numerous simulations. With minimal pre-processing overhead, only a few simulations are needed to construct the ML model, predicting grid-level dynamics with less than 5% error. Unlike ROMs, there is no underlying assumption, ensuring predictions match numerical simulations' granularity without overlooking details.

The study employs numerical reservoir simulation, focusing on steam injection in heavy oil reservoirs. Steam injection is an advanced thermal recovery method used to extract heavy oil and bitumen from deep underground reservoirs. It involves drilling two horizontal wells: a steam injection well and a production well. Steam injection reduces oil viscosity, facilitating gravity-driven flow to the production well. It is primarily used in oil sands reserves like the Athabasca oil sands in Canada but is also suitable for heavy oil reservoirs worldwide.

In the numerical simulation of the steam injection process, several sets of PDEs are solved to model bitumen, water, and steam behavior in the reservoir. These include mass conservation equations for each phase, Darcy's law for fluid flow, heat transfer equations, phase behavior models, relative permeability and capillary pressure models, thermal front tracking, and optional geomechanical effects Butler et al. (1981). The simulator discretizes the reservoir domain, advances in time using iterative techniques, and predicts the steam chamber's movement. These simulations optimize recovery strategies and provide insights into reservoir behavior before field implementation.

The inputs to the simulator are 1/ reservoir properties such as rock type, porosity, and permeability, 2/ fluid Properties such as bitumen viscosity, density, and thermal conductivity, 3/ well configuration such as well depths, spacing, and injection rates, and 4/ operating conditions such as steam temperature, pressure, and duration. The outputs of the simulations are 1/ oil production rates over time, 2/ heat distribution and how heat spreads within the reservoir. Steam is injected to make the viscous bitumen flow, so understanding temperature profiles is crucial, 3/Pressure variations during steam injection and oil production.

## 2   OBJECTIVES

A novel category of neural network architectures has recently emerged, aimed at revolutionizing the approach to learning PDEs Li et al. (2020a;b;c). The fundamental concept driving the development of these architectures is to devise ML algorithms capable of mapping a specific input set to the solution of a PDE. Subsequently, these ML models are employed to predict solutions for the same PDE but under varying initial and boundary conditions or grid resolutions, effectively achieving zero-shot generalization. Essentially, the ML algorithms are engineered to facilitate the learning of PDEs, enabling them to find solutions for a given set of coefficients and initial/boundary conditions. Notably,

these models function as general-purpose solvers, operating by mapping two infinite-dimensional functions through operator learning rather than vector-to-vector learning.

The objective of this work is to employ a deep learning model featuring a neural operator that directly parameterizes the integral kernel in Fourier space to predict the entire steam injection process lifecycle (i.e., the output fields for all the simulation timesteps). That is, by inputting reservoir properties, well configuration, and operating conditions, we obtain grid-level information on oil saturation, production rates, viscosity, and temperature distribution across hundreds of time steps with a fraction of time and computational resources of running numerical simulations with less than 10% prediction error at the grid-level (less than 5% at the aggregated reservoir level). This advancement promises enhanced reservoir management, optimized production, and reduced environmental impact.

## 3   SIMULATION OF THE STEAM INJECTION PROCESS

Numerical simulations of the steam injection process were carried out using CMG STARS, a thermal reservoir simulator capable of handling multiple phases and components Butler et al. (1981).

A 2D heterogeneous model (rock properties (porosity and permeability) in each grid block) was constructed with uniform grid block sizes. Figure 1 illustrates the permeability distribution within the model. Figure 2 illustrates the temperature and oil saturation profiles for the model after 1, 2, 3, and 5 years of steam injection.

## 4   NEURAL NETWORK ARCHITECTURE

Let $D \subset \mathbb{R}^d$ and $A = A(D; \mathbb{R}^{d_a})$ and $U = U(D; \mathbb{R}^{d_u})$ are the input and output function spaces, respectively. The aim is to build a parametric map such that

$$K_\theta : A \longrightarrow U, \quad \theta \in \Theta, \tag{1}$$

where $\Theta$ is the parameter space and $u_j = K_\theta(a_j)$ and $\{a_j, u_j\}_{j=1}^N$ are the set of observations. In order to learn the neural operator $(K_\theta)$, an iterative process $\nu_0 \rightarrow \nu_1 \rightarrow \cdots \rightarrow \nu_n$ is utilized where $\nu_i$ for $i = 1, \ldots, n-1$ are a sequence of functions. In each iteration, $\nu_{i+1}$ is determined from $\nu_i$ by

$$\nu_{i+1}(x) := \sigma \left( W\nu_i(x) + (K(a; \theta)\nu_i)(x) \right), \quad x \in D. \tag{2}$$

In the above equation, which is described in more detail below, $\sigma$ is a non-linear activation function, $W$ is a linear transformation, and $K$ is a non-local integral operator Li et al. (2020a;b;c).

In analogy to the conventional deep neural network (DNN), the above function simply takes a set of input functions ($\{a_j\}_{j=1}^N$) and transforms them from layer $\nu_i$ to $\nu_{i+1}$. There are two main differences between the conventional DNN and the neural operator approach. First, in conventional DNN architectures, a vector-to-vector transform occurs, whereas the neural operator architecture maps two infinite-dimensional functions. The second difference is that an integral kernel operator is used as the transformation tool between two consecutive layers since functions are transferred rather than vectors. The integral kernel operator, akin to the Green's function method to solve PDEs, is defined as

$$(K(a; \theta)\nu_i)(x) = \int_D k\left(x, y, a(x), a(y); \theta\right) \nu_i(y), dy. \tag{3}$$

where $k(\theta)$ is the kernel function that is learned from the data and as an analogy to convolutional neural networks, plays a similar role as the kernel matrix which is applied to an input image. The choice of $k(\theta)$ depends on the problem and several variations including the graph neural operator, multipole graph neural operator, low-rank neural operator, and Fourier neural operator have been implemented in the literature. The Fourier neural operators have been proved to be applicable for solving PDEs such as the Navier-Stokes equation Li et al. (2020a).

The Fourier and inverse Fourier transform of a function is defined as

$$(Ff)_j(k) = \int_D f_j(x)e^{-2i\pi \langle x,k \rangle} dx, \qquad (F^{-1}f)_j(x) = \int_D f_j(k)e^{2i\pi \langle x,k \rangle} dk, \tag{4}$$

where $F$ denotes the Fourier transform of function $f$, $F^{-1}$ denotes the inverse Fourier transform, and $i$ is the imaginary unit. Applying the convolution theorem to the equation:

$$(K(a;\theta)\nu_i)(x) = F^{-1}\left(F(k_\theta) \cdot F(\nu_i)\right) . \qquad (5)$$

This model incorporates a neural operator, which parameterizes the integral kernel directly in Fourier space, enabling DNN to be independent of the mesh. The architecture of the DNN is detailed in (Li et al., 2020a;b;c).

## 5 PROBLEM SET UP

Our proposed methodology involves developing and evaluating an ML model to serve as a surrogate for the numerical simulations for 2D reservoir models. We begin by generating six numerical models, each with randomly altered input parameters, specifically focusing on the rock permeability distribution. For these models, we run numerical simulations to observe their behavior throughout the entire simulation period (400 time steps). The input data and corresponding output fields from five of these models serve as training data for the ML model. By doing so, the ML model learns from the complete evolution of the steam injection process across these different simulations.

To assess the robustness of our model, we compare its predictions with numerical simulation results from a reservoir model that was not part of the training data. This additional reservoir model is constructed using a different random permeability field from those in the training models. During inference, the ML model requires only the first 10 time steps of numerical simulation results and generates the results for the subsequent 40 time steps in an autoregressive manner. That is the last 10 predicted time steps serve as input for the next set of inference steps, and this process continues until predictions are made for the entire desired length (400 time steps). By incorporating this autoregressive technique, we enhance the model's ability to capture temporal dependencies and improve its robustness in predicting the evolution of the steam injection process over time. Finally, we compare the ML model's results with those obtained from numerical simulations at the grid level.

## 6 RESULTS AND DIACUSSIONS

Figures 3, 4, 5, and 6 display a comparison of temperature, pressure, oil saturation, and viscosity predictions obtained from our surrogate models versus numerical simulation results at time steps 100, 200, 300, and 400, respectively. In each figure, the left column illustrates predictions from the ML model, while the right column shows the ground truth (numerical simulation results). Additionally, columns 3 and 4 illustrate the absolute differences between model predictions and numerical simulation results at the grid level, along with the error distribution for each time step.

The ML model demonstrates the capability to predict numerical simulation results at the grid level, spanning from the initial state to the end of a 400-time-step simulation, where 99% of the error values are bounded within 10% error. These results are particularly promising, as our proposed technique drastically reduces the computational cost of numerical simulations, from hours to seconds.

## 7 CONCLUSIONS

Numerical simulations provide valuable insights into fluid flow, heat transfer, and other reservoir behaviors for energy recovery optimization. However, their computational demands can be prohibitive. To tackle this challenge, we propose a deep learning approach that utilizes a neural operator to parameterize the integral kernel in Fourier space. This methodology significantly reduces computational time, enabling real-time reservoir management. The model has been successfully applied to a steam injection use case, accurately predicting simulation outcomes with error rates consistently within 10 percent.

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

# A APPENDIX

## A.1 RESERVOIR MODEL DETAILS

The 2D model dimensions are $26\ m \times 1\ m \times 20\ m$. The model employed a uniform grid block size of $1\ m$ in all directions. An injector well and a producer well were positioned $5\ m$ apart on one side of the model. Table 1 provides a concise summary of the model details, while Figure 1 illustrates the permeability distribution within the model. Figure 2 illustrates the temperature and saturation profiles for the model after 1, 2, 3, and 5 years of steam injection.

A fluid model was constructed using literature data Nourozieh et al. (2015) as a basis. Within the STARS software, the calculation of the partial molar volume of oil is performed by:

$$\nu = \frac{\exp\left(c_{t1}(T - T_R) + \frac{1}{2}c_{t2}(T^2 - T_R^2) - c_p(P - P_R) - c_{pt}(T - T_R)(P - P_R)\right)}{\rho_0} \tag{6}$$

The variable $\rho_0$ denotes the partial mass density at reference pressure $P_R$ and temperature $T_R$. Oil density parameters were calibrated to achieve an Average Absolute Relative Deviation (AARD) of 0.06%. A molecular weight of 539 g/mol was assumed. Typical heavy oil viscosities were used, defined by a temperature-dependent viscosity table in the fluid section.

In this investigation, standard steam relative permeability curves (following Kisman) were employed to characterize the oil-water and gas-liquid relative permeabilities, illustrated in Figure 3. The Stone 2 method was applied to compute the three-phase oil relative permeabilities.

The model begins with a reservoir temperature of 13°C and pressure of 1000 kPa. To simulate the circulation/preheating phase of the steam injection, heater wells are employed for three months to

Table 1: Reservoir and fluid parameters used in the model

| Property | Value |
|---|---|
| Initial reservoir pressure | 1000 kPa |
| Initial reservoir temperature | 13°C |
| Reservoir top | 500 m |
| Reservoir bottom | 520 m |
| Pay thickness | 20 m |
| Formation compressibility | $3 \times 10^{-6}$ 1/kPa |
| Rock volumetric heat capacity | $2.3 \times 10^{6}$ J/(m$^3$.C) |
| Thermal conductivity of rock | $1.61 \times 10^{5}$ J/m-day-C |
| Thermal conductivity of oil | $1.61 \times 10^{5}$ J/m-day-C |
| Thermal conductivity of water | $1.61 \times 10^{5}$ J/m-day-C |
| Porosity | 0.33 |
| Permeability | See figure |
| Initial oil saturation | 0.75 |
| Initial water saturation | 0.25 |
| Thermal diffusivity | $8.1 \times 10^{-7}$ m$^2$/s |
| Oil compressibility (cp in eq. 1) | $3.358 \times 10^{-7}$ 1/kPa |
| Oil thermal expansion coefficient 1 (ct1 in eq. 1) | $4.751 \times 10^{-4}$ 1/K |
| Oil thermal expansion coefficient 2 (ct2 in eq. 1) | $3.81 \times 10^{-7}$ 1/K$^2$ |
| Oil density cross-term (cpt in eq. 1) | $2.29 \times 10^{-9}$ 1/(K.kPa) |
| Steam injected temperature | 215°C |
| Steam injected pressure | 2150 kPa |
| Steam quality | 1.0 |
| Time interval | 1 January 2009 – 1 Nov 2015 |

establish initial communication between injector and producer. During steam injection, injection pressure is maintained at 2150 kPa, and subcooling is used at the producer to prevent live steam production. Steam at 215°C with 100% quality is injected. Figure 2 depicts temperature and saturation profiles after 1, 2, 3, and 5 years of steam injection.

## A.2 FIGURES

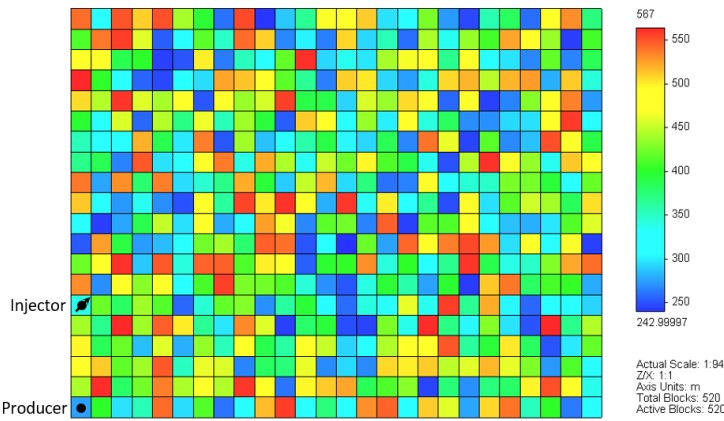

Figure 1: The permeability distribution in the 2D reservoir model.

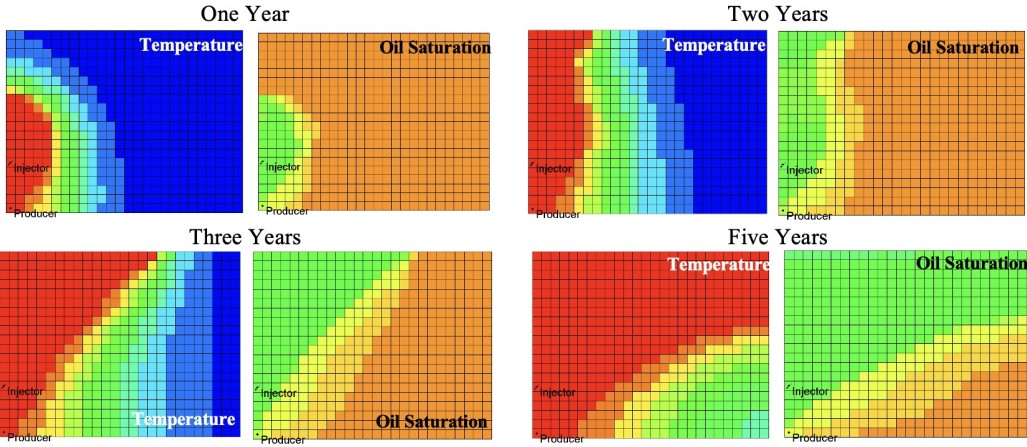

Figure 2: Temperature and oil saturation distributions in the reservoir model at different times.

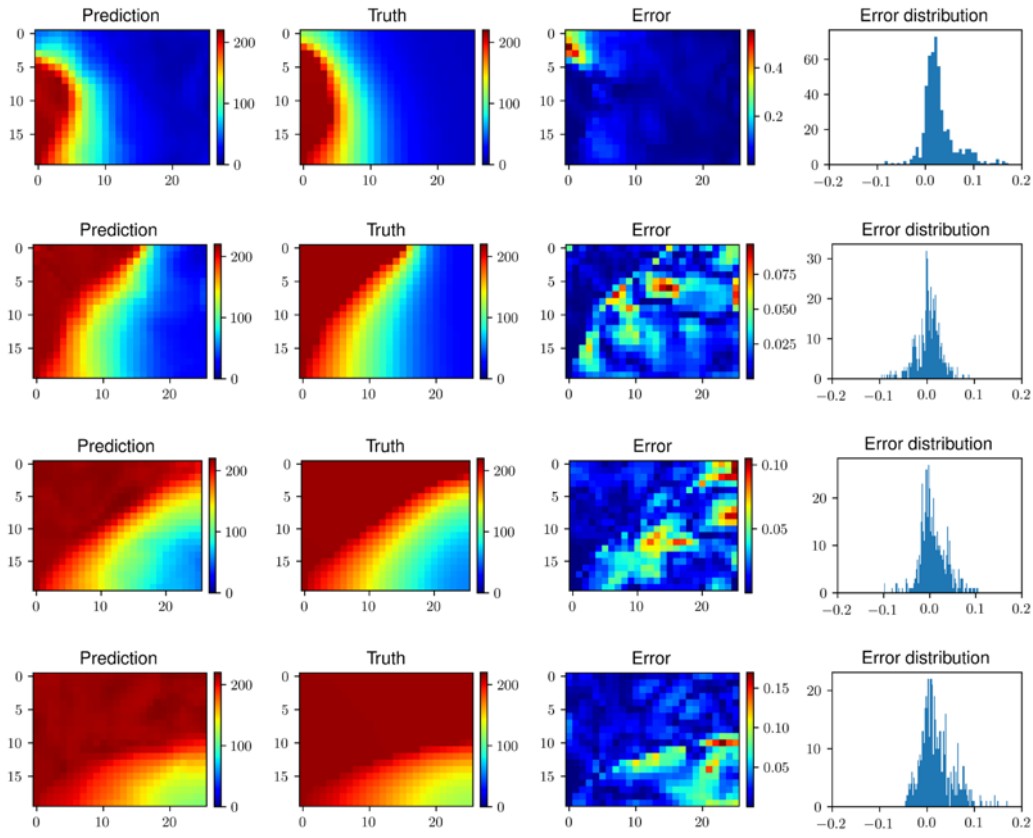

Figure 3: The results of temperature evolution with time for the unseen reservoir model (test model) for time-steps 100, 200, 300, and 400. The first column represents the prediction from our neural network model and the second column represents the results from the numerical simulator.

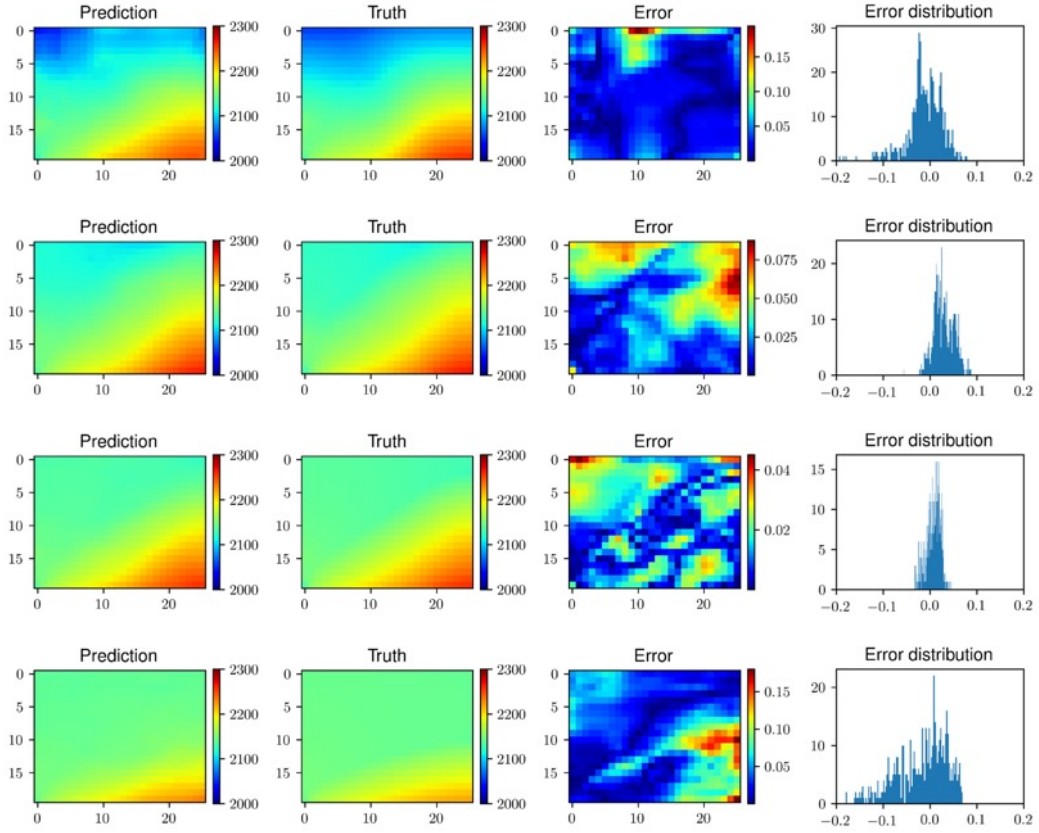

Figure 4: The results of pressure evolution with time for the unseen reservoir model (test model) for time-steps 100, 200, 300, and 400. The first column represents the prediction from our neural network model and the second column represents the results from the numerical simulator.

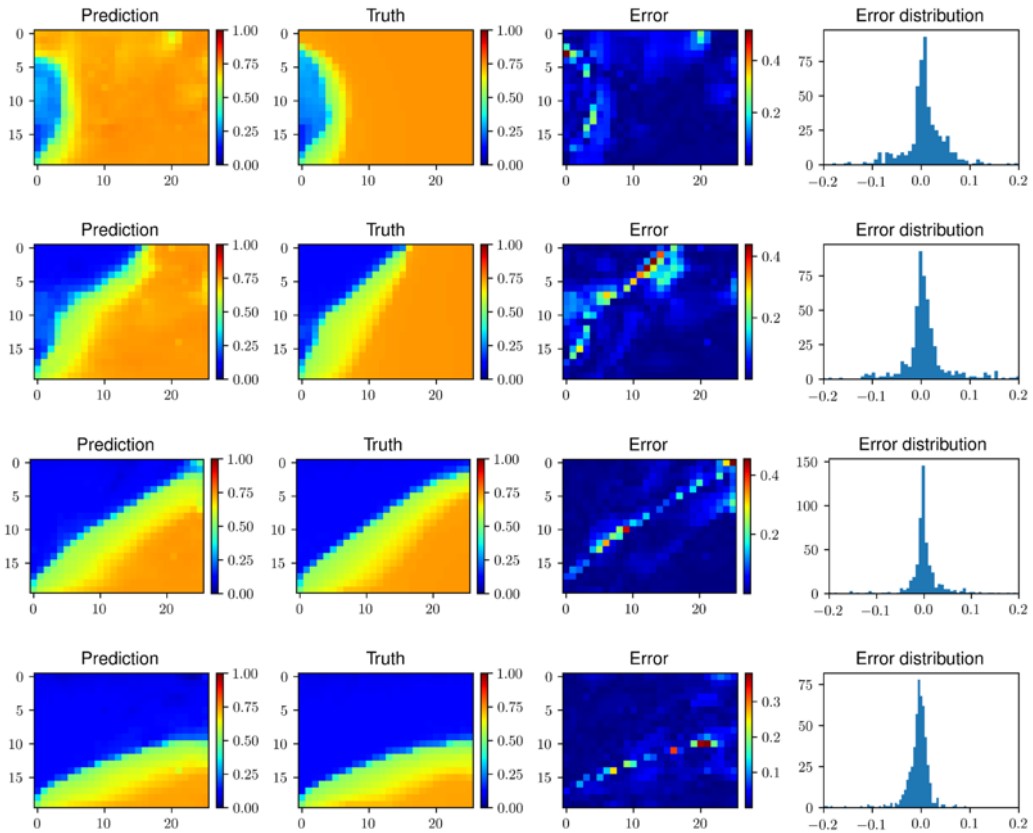

Figure 5: The results of oil saturation evolution with time for the unseen reservoir model (test model) for time-steps 100, 200, 300, and 400. The first column represents the prediction from our neural network model and the second column represents the results from the numerical simulator.

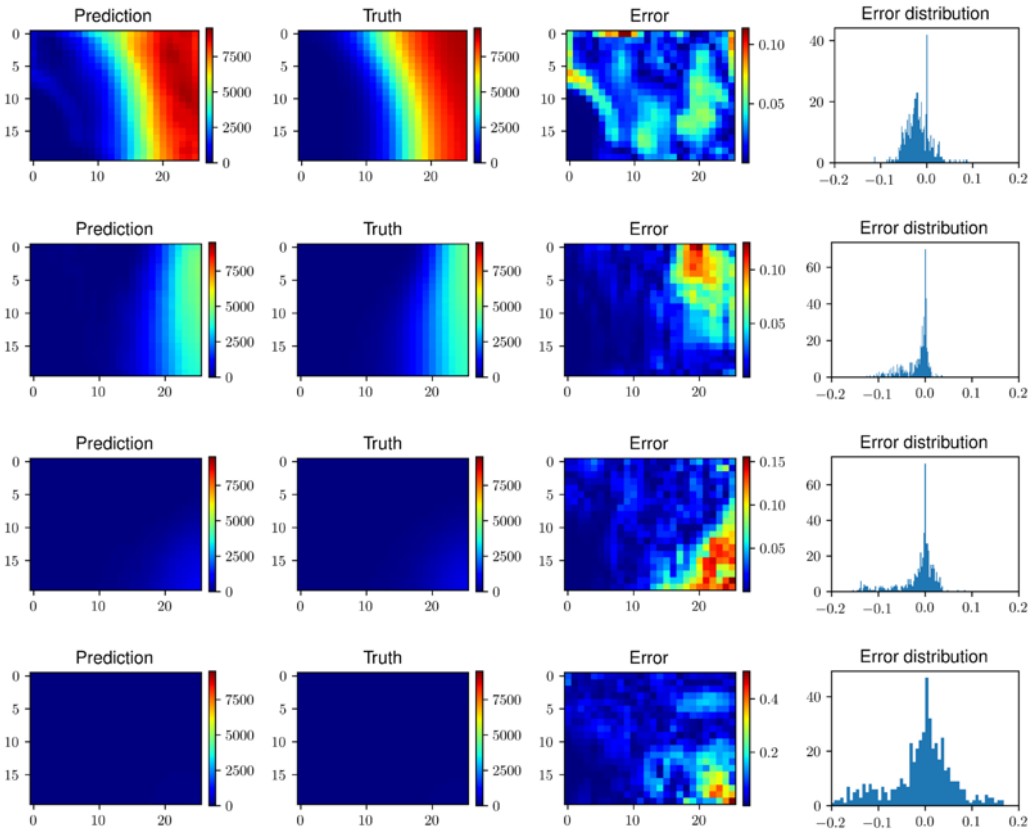

Figure 6: The results of viscosity evolution with time for the unseen reservoir model (test model) for time-steps 100, 200, 300, and 400. The first column represents the prediction from our neural network model and the second column represents the results from the numerical simulator.

