# OpenReview forum: "A Novel ML Model for Numerical Simulations Leveraging Fourier Neural Operators"
_ICLR.cc/2024/Workshop/AI4DiffEqtnsInSci — AI4DiffEqtnsInSci @ ICLR 2024 Poster_

### Official Review · Reviewer_Gm63 · 2024-02-26
**Deep Learning to accelerate reservoir energy management**

**Rating:** 5
**Confidence:** 4

**Review:**

This paper introduces a deep learning methodology to optimize energy recovery in reservoir management by addressing the computational challenges of traditional numerical simulations. The proposed approach utilizes a neural operator in Fourier space to more efficiently predict fluid flow behaviors, significantly reducing computational time from hours to seconds. The authors evaluate the methodology on simulations of steam injection in high-viscosity oil reservoirs.

In general, the present study is of interest for the reservoir energy management and machine learning communities. Although the aim of the paper is clearly defined in the manuscript, it does so repeatedly on different sections. The text can be substantially improved. Particularly, the manuscript can substantially benefit from a clearer structure and logic, as well as avoiding repetitive details. Moreover, the result section clearly lacks a proper description of the results (e.g., highlighting key findings), and the actual analysis is quite limited (e.g., different metrics and diagnostics to evaluate the performance of the ML model would be desired). The theoretical concepts are vaguely presented in the main text. Although further details are described in the appendix, they are not linked with the main text. The figures and the experiments are in general poor, e.g., missing units and confusion between what is a numerical model and different simulations. The results barely support the conclusions (i.e., further analyses and diagnostics are required). Finally, the manuscript lacks a proper discussion to put this work into a broader context, as well as to highlight the limitations of this approach and potential future work.

Specific comments:
- Introduction section. In general, references to support the description of the previous work is missing. Moreover, what are those “handful of ML-based surrogate approaches”?
- Introduction section. The description about the approach of this work does not belong here, and is repeated later on.
- Objective section. This can be a couple of sentences at the end of the introduction, therefore, saving quite same text for better describing the results.
- Simulation of the steam injection process section. Currently the text is misleading. I understand that the authors are using just one physical model (CMG STARS) with 5 different setups and running one simulation for each configuration. If this is the case, there are not five different models to train the ML algorithms, but rather 5 different simulations with specific set of properties. Please, clarify and point to the appendix for a detailed description of the physical model.
- Neural network architecture section. A minimum information about the DNN architecture is required to understand the method, and therefore, the results. Pointing to the original paper is not enough.
- Results and discussion section. This section needs to be extended substantially. By re-writing the former sections and avoiding repeating information, this section should present and discuss (currently is lacking) the results in greater detail. For example, different metrics and diagnostics to evaluate the performance of the ML model would be desired.
- Conclusion section. Although a summary of the results is presented, this section may be missing a discussion about the limitation of the present approach and potential future work.

---

### Official Review · Reviewer_1rf6 · 2024-02-27
**Review of "A Novel ML Model for Numerical Simulations Leveraging Fourier Neural Operators"**

**Rating:** 3
**Confidence:** 4

**Review:**

The manuscript titled "A Novel ML Model for Numerical Simulations Leveraging Fourier Neural Operators" describes the use of Fourier Neural Operators for simulations of reservoir management. The main claim of the paper is the development of a novel neural operator that parameterized the integral kernel directly in Fourier space. This does not seem to be correct since the formulation is exactly the same as Fourier Neural Operators in Li et. al. 2020. The main novelty in the paper seems to be application of FNO to this steam injection problem. This is also something that is very similar to previous work: Wen, Gege, et al. "Accelerating carbon capture and storage modeling using fourier neural operators." arXiv (2022) and Wen, Gege, et al. "Real-time high-resolution CO 2 geological storage prediction using nested Fourier neural operators." Energy & Environmental Science 16.4 (2023): 1732-1741. Due to the lack of significant novelty in the paper, I cannot recommend accepting it. Some more specific comments below:
- Section 1: "There are also a handful of ML-based surrogate approaches proposed in the literature." Cite these works here?
- Section 1: "novel neural operator that parameterizes the integral kernel directly in Fourier space" How is this different from the Fourier Neural Operator (Li et. al. 2020)?
- Eq 2: Each step is just a single FNO layer?
- Section 5: "the ML model requires only the first 10 time steps of numerical simulation results and generates the results for the subsequent 40 time steps in an autoregressive manner." In the previous section, it was mentioned that \nu_{i+1} is predicted based on only \nu_i. How is this history of 10 time steps used then?

---

### Meta-Review · Area_Chair_k2ww · 2024-03-03

**Recommendation:** Accept (Poster)

**Metareview:**

This paper introduces a deep learning methodology to optimize energy recovery in reservoir management by addressing the computational challenges of traditional numerical simulations. The proposed approach utilizes a neural operator in Fourier space to more efficiently predict fluid flow behaviors, significantly reducing computational time from hours to seconds. The authors evaluate the methodology on simulations of steam injection in high-viscosity oil reservoirs. There are several shortcomings in this submission, specifically on the lack of detail on the method itself, the numerical simulations that were used to generate the training dataset and how this specifically differs from other works. Nevertheless this is an interesting contribution to the oil & gas community and for that reason it should be accepted for a poster session.

---

### Decision · Program_Chairs · 2024-03-03

Accept (Poster)